# Rice and Greenhouse Identification in Plateau Areas Incorporating Sentinel-1/2 Optical and Radar Remote Sensing Data from Google Earth Engine

**Tao Zhang** [1], **Bo-Hui Tang** [1,2,*], **Liang Huang** [1,3] **and Guokun Chen** [1]

1   Faculty of Land Resource Engineering, Kunming University of Science and Technology, Kunming 650093, China
2   State Key Laboratory of Resources and Environment Information System, Institute of Geographic Sciences and Natural Resources Research, Chinese Academy of Sciences, Beijing 100101, China
3   Surveying and Mapping Geo-Informatics Technology Research Center on Plateau Mountains of Yunnan Higher Education, Kunming 650093, China
*   Correspondence: tangbh@kust.edu.cn

**Abstract:** Affected by geographical location and climatic conditions, crop classification in the Yunnan Plateau of China is greatly restricted by the low utilization rate of annual optical data, complex crop planting structure, and broken cultivated land. This paper combines monthly Sentinel-2 optical remote sensing data with Sentinel-1 radar data to minimize cloud interference to conduct crop classification for plateau areas. However, pixel classification will inevitably produce a "different spectrum of the same object, foreign objects in the same spectrum". A principal component feature synthesis method is developed for multi-source remote sensing data (PCA-MR) to improve classification accuracy. In order to compare and analyze the classification effect of PCA-MR combined with multi-source remote sensing data, we constructed 11 classification scenarios using the Google Earth Engine platform and random forest algorithm (RF). The results show that: (1) the classification accuracy is 79.98% by using Sentinel-1 data and 91.18% when using Sentinel-2 data. When integrating Sentinel-1 and Sentinel-2 data, the accuracy is 92.31%. By analyzing the influence of texture features on classification under different feature combinations, it was found that optical texture features affected the recognition accuracy of rice to a lesser extent. (2) The errors will be reduced if the PCA-MR feature is involved in the classification, and the classification accuracy and Kappa coefficient are improved to 93.47% and 0.92, respectively.

**Keywords:** classification; Sentinel-1/2; principal component analysis; time series; Google Earth Engine

## 1. Introduction

Timely and accurate access to crop cultivation structure and spatial distribution is conducive to the smooth implementation of precision and smart agriculture and is essential for forecasting food production, prices, and the country's food security. At present, in the process of food security in southwest China, there are still constraints from a lack of cultivated land resources, a large rural population, and a large degree of poverty [1]. Therefore, real-time and dynamic monitoring of crop cover in southwest China is urgently needed. However, due to the complexity of topographic conditions and geographical location, remote sensing monitoring of crops in southwest China has always been a hot spot and difficulty [2].

As one of the three major food crops, rice provides the daily energy source for nearly 50% of the world's population, and 12% of the world's arable land is planted with rice, and this number is increasing year by year. In addition, with the continuous development of agricultural production technology in the past 20 years, the yield of crops has been greatly improved. Greenhouse technology is one of the typical examples [3]. Therefore, the rapid

and accurate acquisition of large-scale mapping of rice and greenhouses has important practical significance for land use transformation and production guidance.

Optical remote sensing data is a key data source in the crop extraction process, and methods for crop identification and monitoring using these data are well established [4–9]. However, due to the influence of clouds and rain, less than 10% of valid optical image data (including less than 20% cloud cover) is available annually in southwest China [10], and the quantity and quality of data cannot be guaranteed. Some scholars have adopted different methods to improve the accuracy of ground object extraction [11–15], but they all need a large amount of optical data support. Sentinel-1 C-band synthetic aperture radar (SAR) images are not affected by clouds or rain and have an all-weather observation capability. The penetrating capability of the radar satellite can obtain vegetation surface information. However, the precision of ground object extraction from single-time SAR data is not good [16]. Combining optical data and radar data to take advantage of active and passive remote sensing can effectively reduce cloud interference and improve the recognition accuracy of land cover in cloudy tropical areas [17–19]. For example, Li et al. used time series optical and radar data to construct different features and obtained mapping results for winter wheat in Henan Province with an overall accuracy of 92.7% [20]. Massimiliano et al. proposed a W-Net method to improve the results of fused Sentinel-1/2 data mapping, which showed suitable performance in distinguishing between rice, water, and bare soil [21]. Steinhausen et al. performed map cover classification of the Chennai basin in order to investigate the classification performance of optical and radar data, and the results showed that the addition of radar data had a positive effect on the identification of rice fields [22]. Tavares et al. selected Sentinel-1/2 data combined with machine learning in order to obtain accurate land cover in the tropics, and the results showed only the lowest classification performance using radar data and the highest performance using combined optical and radar data [23]. While these approaches have demonstrated that combining radar and optical data can improve performance over using a single sensor, they treat the two data sources as completely independent of each other.

Most studies are generally based on single-machine classification, which is difficult to obtain data from, complicated preprocessing, long classification time, and costs in human and financial resources [24,25]. Therefore, cloud-based computing platforms such as Google Earth Engine (GEE) are needed to quickly acquire and process a sufficient number of satellite images. GEE is an open platform that provides satellite observations on a global scale. It has powerful cloud computing capability, can provide a large number of basic data and programming platforms for fast and accurate remote sensing monitoring, and can achieve efficient and accurate large-scale remote sensing mapping [26–28]. Especially in highland areas where optical data is insufficient due to clouds and rain, the GEE platform enables rapid image acquisition and processing to generate cloud-free data. The unexpected outbreak of the pandemic in 2020 led to a global economic downturn and reduced national funding for scientific research. All research institutions and researchers should actively and effectively make rational use of free resources and open platforms.

At present, remote sensing recognition based on the GEE platform mainly uses a single data source in the platform for classification and feature construction, but different data sources have different practicability. GEE has available multi-source data, which can not only reduce the data gap and uncertainty of a single data source but also improve the classification results. Therefore, based on the characteristics of optical and radar data, we propose image fusion and PCA-MR methods to construct a framework for the identification of six land cover types, such as rice and greenhouse, in highland areas, which has three main contributions:

(1)  In the highland region, the optical texture features have less impact on the classification accuracy of images under complex imaging conditions. Specifically, fusing optical and radar data classification as well as using only optical data classification showed that the addition of texture features did not dominate with increasing feature values in complex parcel classification accuracy;

(2) The construction of the PCA-MR method can improve the problem of "different spectrum of the same object, foreign objects in the same spectrum" caused by plot fragmentation and the surrounding environment in the plateau area;

(3) This recognition framework makes full use of GEE multi-source data to simplify the acquisition and processing of data. Theoretically, as long as the phenology information of ground objects is obtained, the mapping results with a resolution of 10 M can be obtained easily and quickly in any test area.

The rest of the paper is structured as follows: Section 2 presents the construction of the data and methods used in this study; Section 3 describes the quantitative and qualitative analysis of the mapping results; Section 4 discusses the results in detail; finally, the conclusion is drawn in Section 5.

## 2. Materials and Methods

### 2.1. Study Area

Luliang County is located in Qujing City, eastern Yunnan Province, China (24°44′–25°18′ N, 103°23′–104°02′ E), with a total area of about 1989.47 square kilometers. The county is located in the plateau area (above 1000 m), belonging to the subtropical plateau monsoon-type wet summer and dry winter climate, with no hot summer, no cold winter, warm and dry spring, and cool and wet autumn characteristics. The annual average temperature is 14.7 °C, and the annual rainfall is 900–1000 mm, with 249 frost-free days and 2442.5 annual sunshine duration hours. Its annual solar radiation is 125.2 kcal/cm$^2$. According to the survey of Yunnan Statistical Yearbook 2020, the output of rice in Luliang County ranks first among all counties in the province, and the main crops are rice and greenhouses vegetables. The study area is shown in Figure 1.

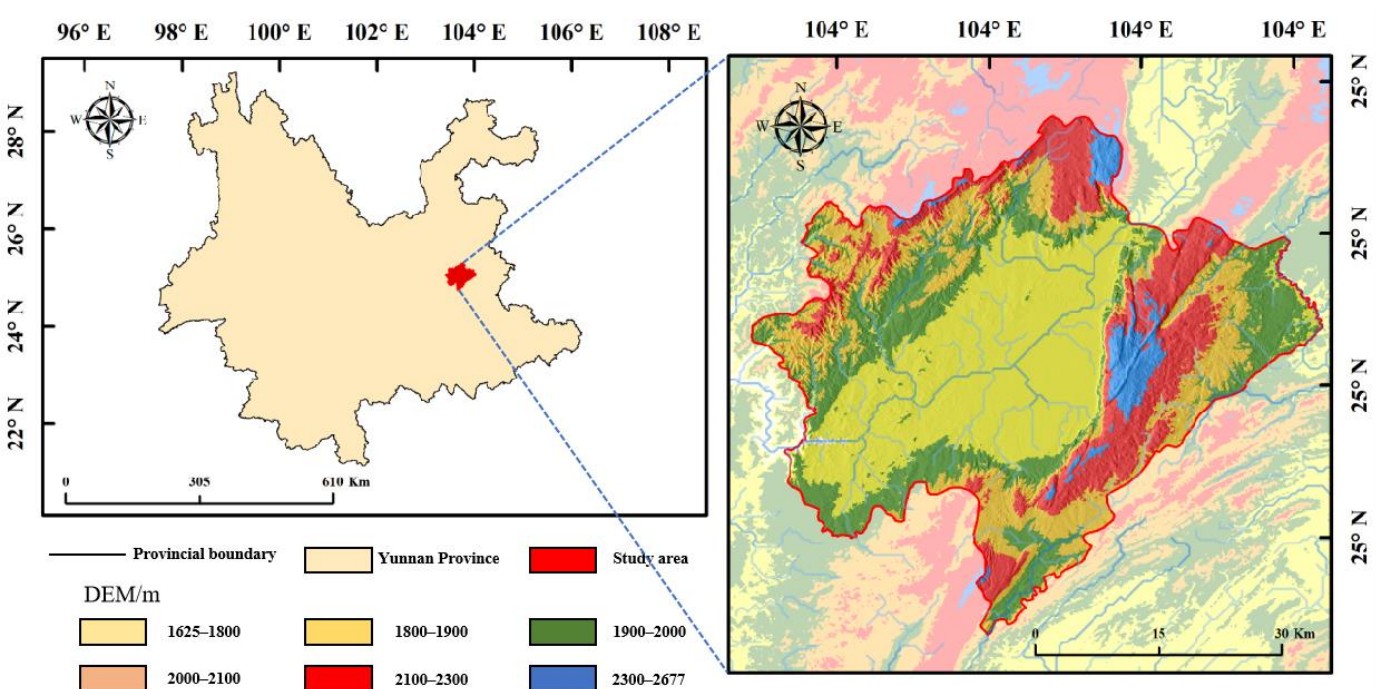

**Figure 1.** Location of the study area. The vector data and DEM data in the figure were downloaded from the Data Center for Resources and Environmental Sciences, Chinese Academy of Sciences (http://www.resdc.cn (accessed on 3 February 2022)).

### 2.2. Data

#### 2.2.1. Sentinel Data

Sentinel-1 SAR (hereinafter referred to as "S1") has undergone orbit recovery, noise reduction, radiometric calibration, and terrain correction. It consists of two polar-orbiting

satellites, A and B, with a temporal resolution of 6 d and a spatial resolution of 10 m. The polarization is the backscattering coefficients of VV and VH. Sentinel-2 MSI (hereinafter referred to as "S2") consists of two high-resolution satellites, 2A and 2B, with a temporal resolution of 5 d and a spatial resolution of 10 m. Sentinel-2 is divided into L1C and L2A level data according to different atmospheric correction states, and L2A level data has been atmospherically corrected, so L2A level data is selected. The Sentinel data contains three QA bands, one of them (QA60) with cloud mask information. Lu Liang County is one of the representative areas of the southwest highlands of China. In this region, rice is sown every year in May–June and harvested from the end of October to the beginning of November. It does not take phenological information into consideration when planning agricultural greenhouses. Therefore, we selected optical and radar data from July–November 2020 and used the ee.filterDate () and ee.filterBounds () codes to perform temporal and study area filtering. Finally, 87 optical and 76 radar data sets were obtained. Then the data with less than 10% cloud amount is filtered, and cloud masking is performed using the QA60 band. Finally, the acquired optical data is normalized using median synthesis to minimize the effect of shadows and clouds. In this study, median synthesis was performed using the code median () on GEE.

### 2.2.2. Reference Data

Combining Sentinel 10 m imagery, the *2020 Yunnan Provincial Statistical Yearbook,* and the Global 30 m Fine Land Cover Data 2020 (GLC_FCS30-2020) provided by the Earth Big Data Shared Service Platform (http://data.casearth.cn/ (accessed on 3 February 2022)), we divided the study area into six categories. The classification categories are: rice, agricultural greenhouses, impervious water surface, water, forest land, and others. The impervious water surface includes construction land such as houses and roads, and others include bare soil, melons fruits, wasteland, and quarries. To ensure the accuracy of the samples, we used Google Earth contemporaneous high-resolution images (resolution of 0.3–2 m), PCA-MR visualization images, and the GLC_FCS30-2020 product for point-by-point selection. The GLC_FCS30-2020 product was used to facilitate point selection against the relative positions of the six categories. PCA-MR visualization images are PCA-processed images of sentinel data (bands: PCA-M1/PCA-M2/PCA-M3), which mainly provide a reference for sample point selection. Each sample point was marked by a manual visual interpretation for a total of 5366 samples. The GEE platform was used to randomly divide all sample points into a training sample (in-bag data) and a test sample (out-of-bag data), both of which have essentially the same amount of data (Table 1). Finally, out-of-bag data are used to evaluate the error, and the obtained error (out-of-bag error) is used to evaluate the accuracy.

**Table 1.** Sample data set of the study area.

| Class | Rice | Greenhouses | Impervious | Water | Forest | Others | Total |
|-------|------|-------------|------------|-------|--------|--------|-------|
| Train | 579 | 588 | 529 | 49 | 255 | 517 | 2517 |
| Test | 643 | 561 | 605 | 54 | 233 | 753 | 2849 |
| Total | 1222 | 1149 | 1134 | 103 | 488 | 1270 | 5366 |

### 2.3. Methods

Figure 2 shows the flowchart of the research technical route in this work, which mainly includes image processing, feature construction, a feature synthesis method based on multi-source remote sensing data, scenario design, and accuracy assessment.

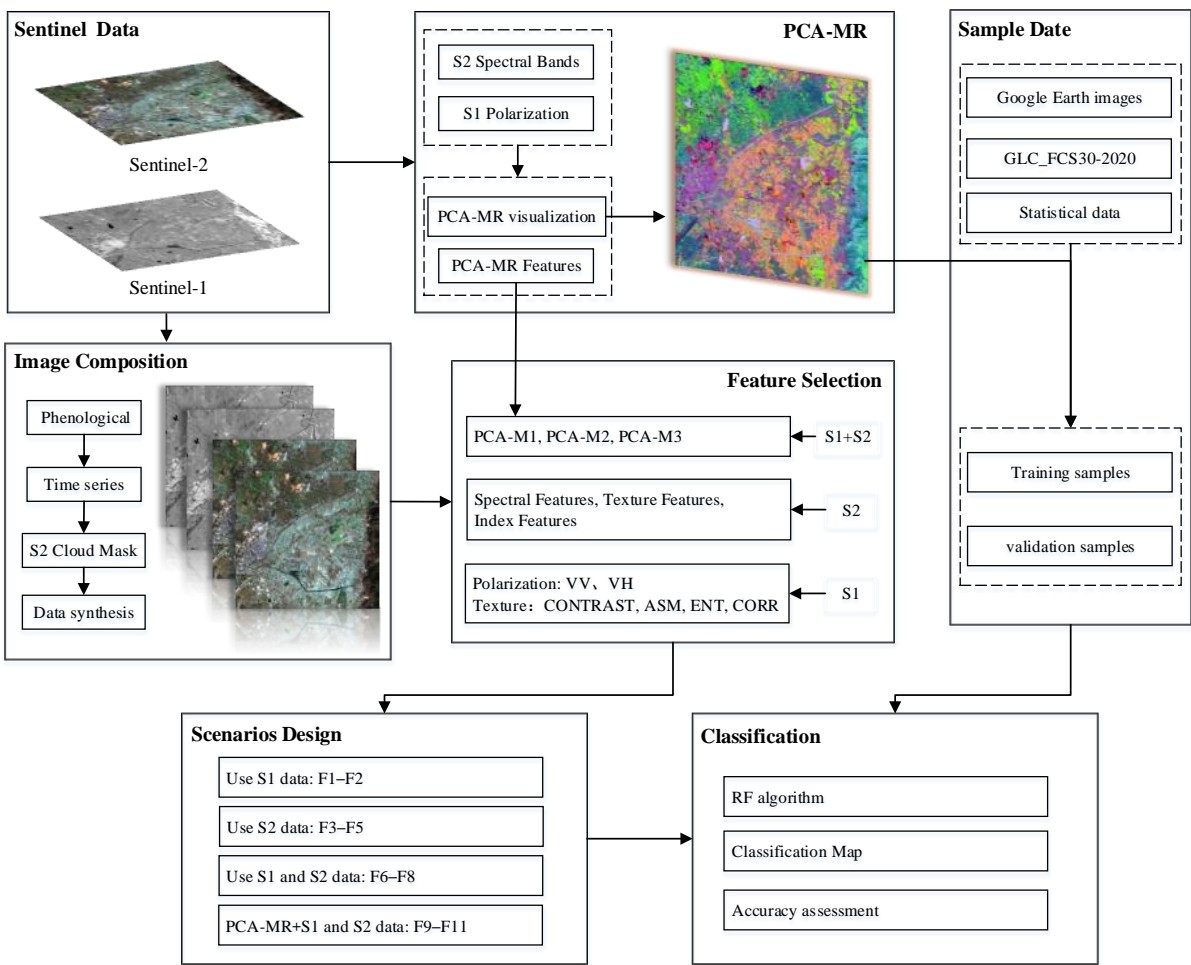

**Figure 2.** Flowchart of the proposed classification framework. S1 and S2 represent Sentinel-1 and Sentinel-2 data, respectively.

### 2.3.1. Random Forest Algorithm

The random forest algorithm is constructed based on the decision tree, which not only contains the sample disturbance idea of the Bagging algorithm but also introduces the attribute disturbance into each node of the decision tree. The number of decision trees has an important effect on the efficiency and accuracy of the RF algorithm. When the number of decision trees is small, the classification accuracy is low. When the number of decision trees is large, the classification accuracy tends to be gradually balanced, but the operation speed will be affected. Previous studies have shown that when the number of decision trees increases, the out-of-pocket errors tend to be stable, and it is meaningless to continue to increase the number of decisions. Therefore, the number of decisions selected in this paper is 100 [29].

### 2.3.2. Principal Component Analysis of Multi-Source Remote Sensing Data (PCA-MR)

(1)　Feature establishment

In combination with the phenological conditions and ecological environment of the study area, 6 features were derived by combining multi-source remote sensing data, with a total of 37 variables (Table 2), including spectral feature, index feature, texture feature, polarization feature and principal component feature after PCA-MR. The index features include: Normalized Difference Vegetation Index (NDVI) [30], Normalized Difference Water Index (NDWI) [31], Enhanced Vegetation Index (EVI) [32], Modified Normalized Difference Water Index (MNDWI) [33], Land Surface Water Index (LSWI) [34], Normalized Difference Built-Up Index (NDBI) [35], Soil-adjusted Vegetation Index (SAVI) [36], and

Green Chlorophyll Vegetation Index (GCVI) [37]. Vegetation Indices can capture the changes of greenness and greenness in the growing period of rice, water, and soil indices can help to distinguish water content information between agricultural greenhouses and other land features. Considering the characteristics of complex planting structure and broken fragmentation in the test area, adding texture features can better describe the randomness and details of the texture, including gray level co-occurrence matrix (GLCM) calculation: contrast (CONTRAST), angular second moment (ASM), correlation (CORR), and entropy (ENT). The calculation formula for texture features is as follows:

$$CONTRAST = \sum_{i=1}^{n} \sum_{j=1}^{n} p(i,j)(i-j)^2 \tag{1}$$

$$ASM = \sum_{i=1}^{n} \sum_{j=1}^{n} p(i,j)^2 \tag{2}$$

$$ENT = -\sum_{i=1}^{n} \sum_{j=1}^{n} p(i,j) \log_2 p(i,j) \tag{3}$$

$$mean = \sum_{i=1}^{n} \sum_{j=1}^{n} p(i,j) \tag{4}$$

$$variance = \sum_{i=1}^{n} \sum_{j=1}^{n} p(i,j)(i-mean) \tag{5}$$

$$CORR = \sum_{i=1}^{n} \sum_{j=1}^{n} (i-mean)(j-mean)\frac{p(i,j)^2}{variance} \tag{6}$$

where $i, j$ are the determinant coordinates of the pixel in the image, $P(i,j)$ is the grayscale joint probability matrix, and $n$ represents the order of the grayscale symbiosis matrix.

**Table 2.** Feature space categories and variables. S1 and S2 represent Sentinel-1 and Sentinel-2 data, respectively.

| Sensor Used | Feature Space | Characteristic Variable | Total |
|---|---|---|---|
| S2 | Spectral | B1, B2, B3, B4, B5, B6, B7, B8, B8A, B9, B11, B12 | 12 |
| | Index | NDVI, NDWI, EVI, LSWI, NDBI, GCVI, SAVI, MNDWI | 8 |
| | Texture | CONTRAST, ASM, ENT, CORR | 4 |
| S1 | Polarization | VV, VH | 2 |
| | Texture (VV, VH) | CONTRAST, ASM, ENT, CORR | 8 |
| S1 + S2 | PCA-MR | PCA-M1, PCA-M2, PCA-M3 | 3 |

(2)   Calculation of PCA-MR

In order to improve the classification efficiency of rice and agricultural greenhouses and improve the errors and impacts caused by pixel classification, we constructed a feature synthesis method (PCA-MR) based on principal component analysis.

The main steps shown in Figure 3 are as follows: (1) median values of S1 and S2 time series data were synthesized based on monthly scales, and S2 data were processed by cloud mask. Then, the cat () function was used in GEE to fuse the images after median values were synthesized to obtain fusion images of the study area. (2) The optical and radar data of the Sentinel have 12 and 2 raw bands, respectively. We first use the 12 bands from the optical data for classification, derive the GINI index [38] weight values by the explain () function in GEE, and rank them. (3) The two low-weighted bands are removed, and the polarization information of Sentinel 1 and the band information of Sentinel 2 are

combined to make a total of 12 bands for principal component analysis, visual display, and feature construction.

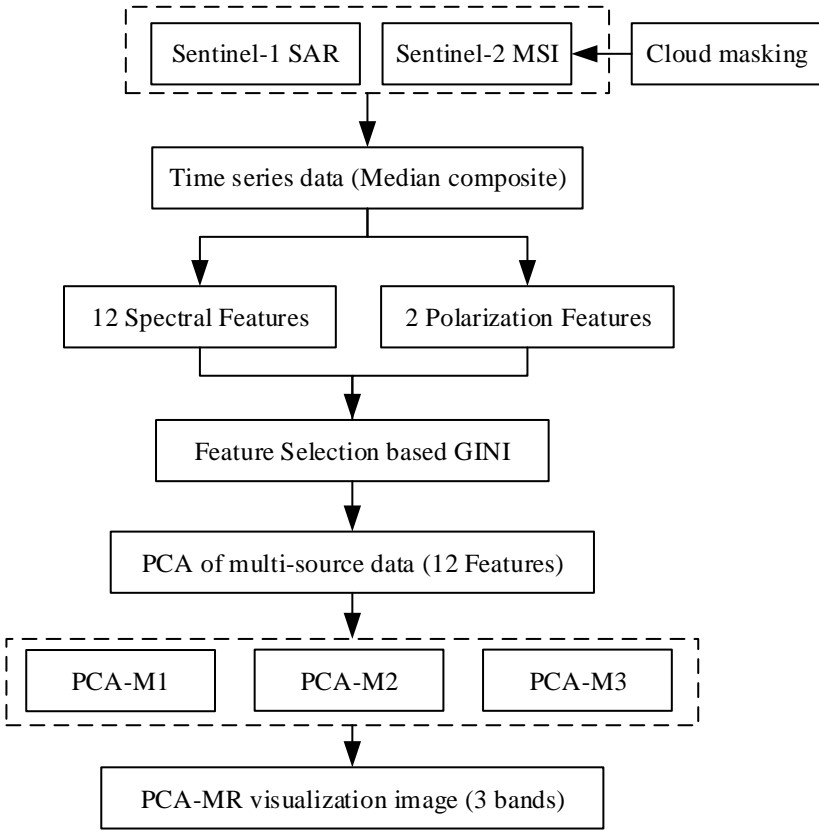

**Figure 3.** Flowchart of the proposed PCA-MR method.

Conventional methods are generally based on a single sensor for principal component analysis. We emphasize the importance of classification feature optimization and joint multi-source remote sensing data in principal component analysis, and all calculations are implemented in GEE platform coding. In general, the first principal component contains more than 80% of ground object variance information of all bands, and the first three principal components can contain about 95% of ground object variance information. Therefore, the first three principal components (PCA-M1, PCA-M2, PCA-M3) after PCA-MR are selected in this paper to distinguish the differences between rice, agricultural greenhouses, and other ground objects.

### 2.3.3. Scenarios Design

In order to evaluate the extraction accuracy and mapping effect of different sensors on rice and agricultural greenhouses in the experimental area, a total of 11 groups of feature scenes were designed (Table 3). Scenario F1 uses S1 data (VV and VH) synthesized at a monthly scale to plot rice and agricultural greenhouses; F3 is constructed from 12 S2 bands (spectral features); F2 and F5 are used to evaluate the contribution of texture features in the classification process; F4 aims to compare the classification performance with and without spectral indices; F6 integrates Sentinel active and passive remote sensing data to explore whether incorporating optical and radar data can improve the accuracy; based on F6, F7 adds index features, aiming to highlight the influence of two kinds of data fusion on index features; F8 is to evaluate the effect of S2 texture features on classification details; scenarios F9-F11 extract principal component features from S1 and S2 fusion images to participate in the classification, comparing them with F6-F8 classification results, and taking the advantages and functions of PCA-MR in the crop extraction process of active and passive remote sensing fusion data.

**Table 3.** The combinations of classification features. S1 and S2 represent Sentinel-1 and Sentinel-2 data, respectively.

| Scenario | Category of the Features | Number of Variables |
|:---:|:---:|:---:|
| F1 | S1 Polarization | 2 |
| F2 | S1(Polarization + Texture) | 10 |
| F3 | S2 Spectral | 12 |
| F4 | S2 (Spectral + Index) | 20 |
| F5 | S2 (Spectral + Index + GLCM) | 24 |
| F6 | S1(Polarization + Texture) + S2 Spectral | 22 |
| F7 | S1(Polarization + Texture) + S2 (Spectral + Index) | 30 |
| F8 | S1(Polarization + Texture) + S2 (Spectral + Index + GLCM) | 34 |
| F9 | S1(Polarization + Texture) + S2 Spectral + PCA-MR | 25 |
| F10 | S1(Polarization + Texture) + S2 (Spectral + Index) + PCA-MR | 33 |
| F11 | S1(Polarization + Texture) + S2 (Spectral + Index + GLCM) + PCA-MR | 37 |

2.3.4. Accuracy Assessment

We used the commonly used statistical methods based on the confusion matrix and kept cross-validation to evaluate the classification accuracy. The specific evaluation metrics include Overall Accuracy (OA), User Accuracy (UA), Producer Accuracy (PA), and Kappa, which are calculated as follows:

$$PA = \frac{n_{ii}}{n_{\cdot i}} \times 100\% \tag{7}$$

$$UA = \frac{n_{ii}}{n_{i\cdot}} \times 100\% \tag{8}$$

$$OA = \frac{\sum\limits_{i=1}^{q} n_{ii}}{n} \times 100\% \tag{9}$$

$$Kappa = \frac{\left[ n \cdot \sum\limits_{i=1}^{q} n_{ii} - \sum\limits_{i=1}^{q} (n_{i\cdot} \cdot n_{\cdot i}) \right]}{\left[ n^2 - \sum\limits_{i=1}^{q} (n_{i\cdot} \cdot n_{\cdot i}) \right]} \tag{10}$$

where $n_{ii}$ is the value of row $i$ and column $i$ in the confusion matrix; $n_{i\cdot}$ is the sum of the $i$ row in the confusion matrix; $n_{\cdot i}$ is the sum of the $i$ column in the confusion matrix; $n$ is the total number of verification samples; $q$ is the total number of columns of the confusion matrix.

**3. Results**

*3.1. Classification with Active and Passive Remote Sensing Data*

3.1.1. Classification with Single Sensor Data

In order to check the classification results of random forests in different scenarios, we conducted repeated experiments and conducted strict accuracy evaluation and comparison (Figure 4 and Table 4). The total accuracy of polarization feature (F1) classification was 76.35%, and Kappa was 0.70. When using GLCM to calculate texture features, fields 4, 8, and 16 were selected for the calculation of the gray co-occurrence matrix, and finally, the four fields with the highest accuracy were selected. After the fusion of S1 texture features (F2), the total accuracy was improved by 3.63%, and Kappa increased to 0.75, indicating that S1 texture features played a positive role in the accurate extraction of F2 crops. When S2 data is used for classification, the total accuracy of the three feature combination scenes is above

90%, among which the total accuracy of F5 (spectral features + index features + S2 texture features) is 91.18%, and Kappa is 0.88, which is 0.52% and 0.32% higher than F3 and F4, respectively. It shows that spectral features play an important role in image classification.

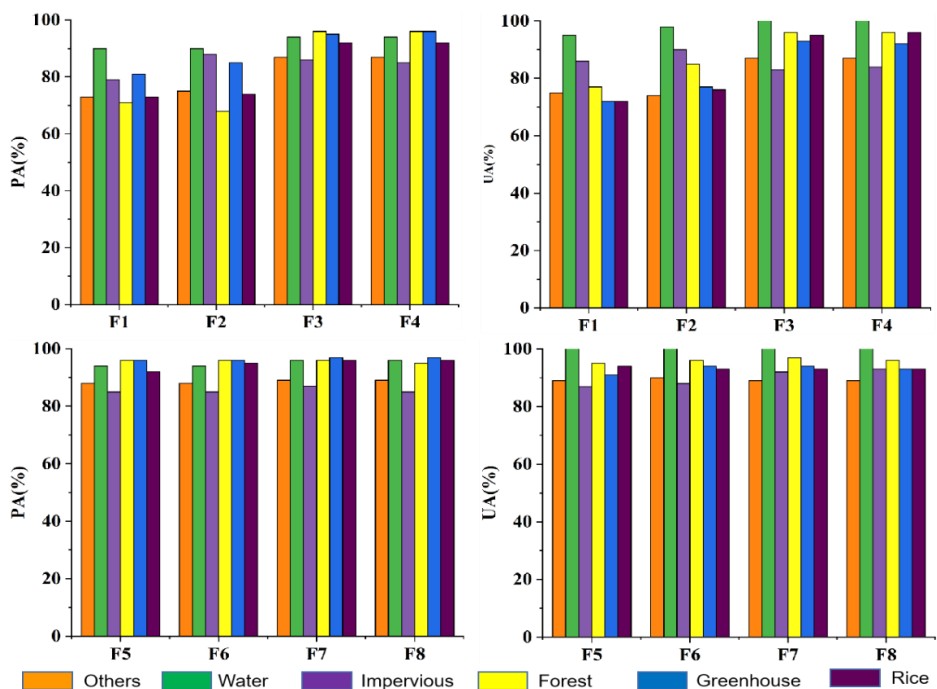

**Figure 4.** PA and UA were compared and evaluated for each classification scenario. F1–F5 is the classification effect of a single data source, and F6–F8 are the classification effect of the fusion of optical and radar data. UA and PA represent the accuracy of users and producers, respectively.

**Table 4.** Comparison of classification accuracy based on points in scenes F1–F4.

| Classes | F1 | | F2 | | F3 | | F4 | |
|---|---|---|---|---|---|---|---|---|
| | PA | UA | PA | UA | PA | UA | PA | UA |
| Rice | 0.81 | 0.72 | 0.85 | 0.77 | 0.95 | 0.93 | 0.96 | 0.92 |
| Greenhouses | 0.73 | 0.72 | 0.74 | 0.76 | 0.92 | 0.95 | 0.92 | 0.96 |
| Impervious | 0.79 | 0.86 | 0.88 | 0.90 | 0.85 | 0.83 | 0.85 | 0.84 |
| Forest | 0.71 | 0.77 | 0.68 | 0.95 | 0.96 | 0.96 | 0.96 | 0.96 |
| Water | 0.90 | 0.95 | 0.90 | 0.98 | 0.94 | 1.00 | 0.94 | 1.00 |
| Others | 0.73 | 0.75 | 0.75 | 0.74 | 0.87 | 0.87 | 0.87 | 0.87 |
| OA (%) | 73.35 | | 79.98 | | 90.66 | | 90.86 | |
| Kappa | 0.70 | | 0.74 | | 0.88 | | 0.88 | |

UA: user's accuracy, OA: overall accuracy, PA: producer's accuracy.

### 3.1.2. Classification with Combined Optical and Radar Data

The combination of optical and radar data can complement each other in the classification process. The total accuracies of scene F6, F7, and F8 classification are more than 92%, among which the total accuracy of F7 is 92.31%, 1.13% higher than that of F5, indicating that the combined S1 and S2 multi-source remote sensing data classification has great improvement and obvious advantages compared with the classification of single sensor data source, which is consistent with the existing research results [20–23]. After the introduction of S2 texture features, the total accuracy of F8 reached 92.45%, only improved by 0.14%, as shown in Figure 5. Comparing the fragment regions of F7 and F8, it was found that when

the region was planted with complex or mixed pixels, the partial classification results of F8 were fragmented and patchy, with no obvious details. This may be caused by the use of adjacent pixels in the calculation of texture features by the gray level co-occurrence matrix and the irregular changes of block broken pixels.

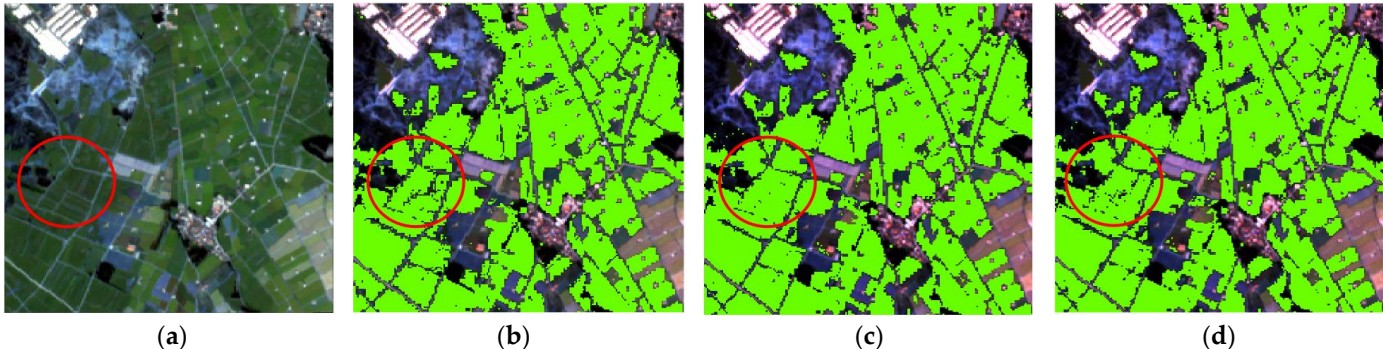

**Figure 5.** Comparison of rice classification results under different feature combinations. (**a**) shows the monthly composite image of S2 data (bands: RED/GREEN/BLUE); (**b**–**d**) show the rice classification results of F3, F7, and F8, respectively.

As for Scenarios F1–F8 crop classification combining S1 and S2 multi-source remote sensing data, scene F7 has the best effect; that is, spectral features + index features + polarization features + radar texture features is the best combination. The paddy field plots in the whole region are broken and irregular in shape, and there are few plots with complete shape. As the sowing time is difficult to unify, the geological period of different plots is slightly different, which is easy to be confused with surrounding forest land and other crops. Therefore, it is difficult for rice to be accurately identified by high-resolution images with a single sensor. The fusion of active and passive remote sensing data classification based on multiple time series S1 and S2 can not only solve the problem of few available images due to more clouds and fogs in plateau and mountainous areas but also enable rice to be accurately classified by adding different features. In general, the PA and UA of rice and agricultural greenhouses are all above 0.9 (Table 5), but the classification results of broken areas are prone to the problems of "different spectrum of the same object, foreign objects in the same spectrum". In order to solve the problem mentioned above, PCA-MR determines and combines three principal components to improve the separability of ground objects.

**Table 5.** Comparison of classification accuracy based on points in scenes F5–F8.

| Classes | F5 | | F6 | | F7 | | F8 | |
|---|---|---|---|---|---|---|---|---|
| | PA | UA | PA | UA | PA | UA | PA | UA |
| Rice | 0.96 | 0.91 | 0.97 | 0.94 | 0.97 | 0.94 | 0.97 | 0.93 |
| Greenhouses | 0.92 | 0.94 | 0.95 | 0.93 | 0.96 | 0.93 | 0.96 | 0.93 |
| Impervious | 0.85 | 0.87 | 0.87 | 0.88 | 0.87 | 0.92 | 0.85 | 0.93 |
| Forest | 0.96 | 0.95 | 0.96 | 0.96 | 0.96 | 0.97 | 0.95 | 0.96 |
| Water | 0.94 | 1.00 | 0.94 | 1.00 | 0.96 | 1.00 | 0.96 | 1.00 |
| Others | 0.88 | 0.89 | 0.88 | 0.90 | 0.88 | 0.89 | 0.89 | 0.89 |
| OA (%) | 91.18 | | 91.90 | | 92.31 | | 92.45 | |
| Kappa | 0.88 | | 0.89 | | 0.90 | | 0.90 | |

UA: user's accuracy, OA: overall accuracy, PA: producer's accuracy.

### 3.2. Classification with Combined PCA-MR and Active and Passive Remote Sensing Data

Considering the error and influence of pixel-based classification, we introduce principal component information to improve the classification effect. Principal component analysis is a kind of feature extraction, in essence, which mainly preserves the main ground object information of the original space in the feature space by dimensionality reduction in data. The dimension of the new feature space is much lower than that of the original space. On the premise of not reducing the "effective" ground object information, the "effective" information with fewer dimensions is used to identify the original data set so as to achieve the purpose of optimal variance and improve the data identification effect [39].

Through repeated experiments, it is found that the principal component analysis method (PCA-MR) constructed by us has great research potential in remote sensing crop extraction. The planting structure in the Yunnan Plateau is complex, the plots are broken, and the differences and connections between different categories are highlighted through PCA-MR; PCA-MR visualization was used to verify and supplement sample points, and the final results were compared and analyzed; through PCA-MR feature combination experiment, the improvement in cartographic accuracy by principal component features under complex imaging conditions was explored. In scenario F3, the 12 original spectral bands of S2 data are used for classification, and the characteristic contribution degree calculated based on the Gini Index shows (Figure 6a) that bands such as B6 and B7 have low contribution degrees and little significance in classification calculation.

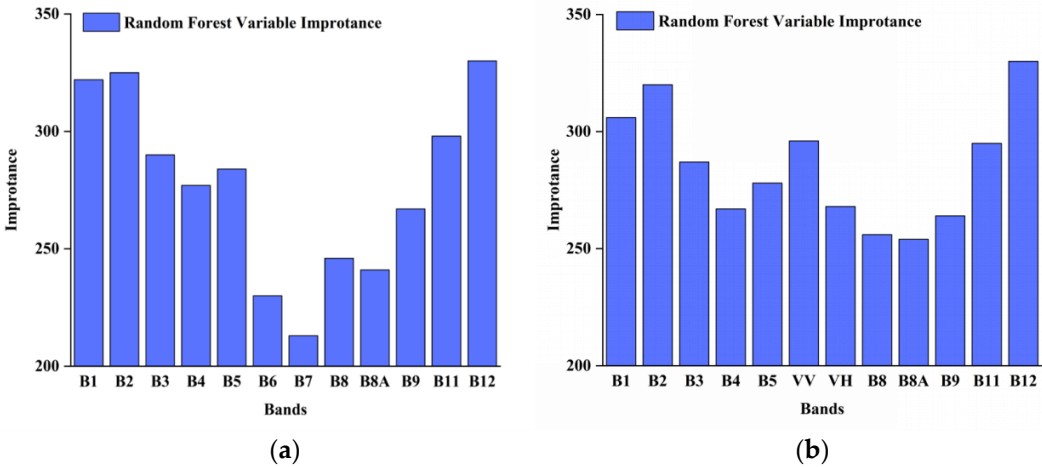

**Figure 6.** Feature weights calculated based on the Gini index. (**a**) shows the importance of using 12 spectral features. (**b**) shows the weight changes of each feature by using VV and VH bands in place of optical data B6 and B7 bands.

The mathematical expression of the Gini Index is as follows:

$$Gini = 1 - \sum_{j}^{J} p^2(j/h) \tag{11}$$

$$p(j/h) = \frac{n_j(h)}{n(h)}, \sum_{j}^{J} p(j/h) = 1 \tag{12}$$

where $P(j/h)$ represents the probability that the test variable $h$ belongs to the $j$th class when samples are randomly selected from the sample set; $n_j(h)$ represents the number of samples that test variable $h$ belongs to the $j$th class; $n(h)$ represents the total number of training samples under test variable $h$; $j$ is for class number.

The fusion data S1 + S2 contains 14 bands (spectral bands + polarization information). If 14 bands are used for operation, the operation speed will be reduced, and data redundancy will be increased. Therefore, B6 and B7 bands are removed, and VV and VH

radar polarization information is introduced. The results show that the distribution of each feature contribution degree is more uniform after the introduction of VV and VH information (Figure 6b).

The classification of rice and agricultural greenhouses by combining PCA-MR and active and passive remote sensing data shows that: (1) PCA-MR visualization can well highlight the ground feature information (Figure 7c) and can distinguish some ground features only by visual interpretation; (2) when F3 and F7 were classified (Figure 7c,e), some houses were misclassified as agricultural greenhouses. When F10 was used for classification, the effect was improved, and the problem of missing and misclassifying was solved to some extent. The reason is that PCA-MR enhanced ground object information and improved separability; (3) the total accuracy of F10 is 93.47%, and Kappa is 0.92, which is 2.81% and 1.16% higher than that of F3 and F7, respectively, indicating that the proposed method has achieved success in the classification of rice and agricultural greenhouses under the condition of land fragmentation and lack of optical data in the plateau region.

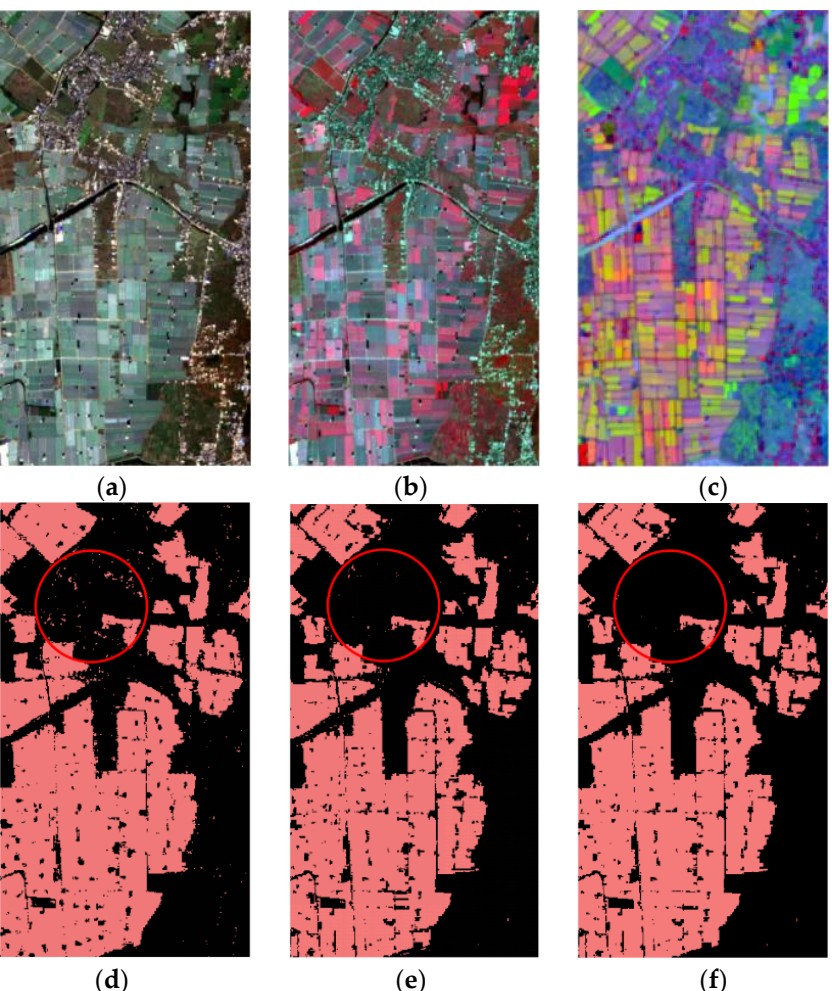

**Figure 7.** Comparison of greenhouse classification results under different character combinations. (**a**–**c**) are, respectively, the visualization results of the S2 image (bands: RED/GREEN/BLUE), S2 false color image (bands: NIR/RED/GREE), and PCA-MR image (bands: PCA-M1/PCA-M2/PCA-M3). (**d**–**f**) Display the classification results of agricultural greenhouses in Scenarios F3, F7, and F10.

After the introduction of PCA-MR information in Scenario F9, F10, and F11, the classification accuracy is shown in Table 6, and the total accuracy of the three scenes is above 92%. Among them, the total accuracy of F11 was 93.40%, which is reduced by 0.07% compared with the accuracy of F10, which again proves that S2 texture features do not contribute much in the classification process and the details of classification results are

not obvious. S2 texture features were added to the region with complex land fragmentation and cover types for classification, and the classification accuracy did not increase with the increase in the number of features. According to the analysis of the classification accuracy of rice and agricultural greenhouses, The UA and PA of F10 ranged from 0.94 to 0.97, showing a suitable mapping effect and complete classification details (Figure 8). Therefore, we determined the best classification scene was F10, that is, spectral features + index features + polarization features +S1 texture features +PCA-MR features.

**Table 6.** Mapping accuracy evaluation of rice and agricultural greenhouses.

|  | F9 | | F10 | | F11 | |
|---|---|---|---|---|---|---|
|  | **Rice** | **Greenhouses** | **Rice** | **Greenhouses** | **Rice** | **Greenhouses** |
| PA | 0.97 | 0.95 | 0.97 | 0.94 | 0.97 | 0.95 |
| UA | 0.94 | 0.96 | 0.96 | 0.96 | 0.93 | 0.96 |
| OA (%) | 92.91 | | 93.47 | | 93.40 | |
| Kappa | 0.91 | | 0.92 | | 0.92 | |

UA: user's accuracy, OA: overall accuracy, PA: producer's accuracy.

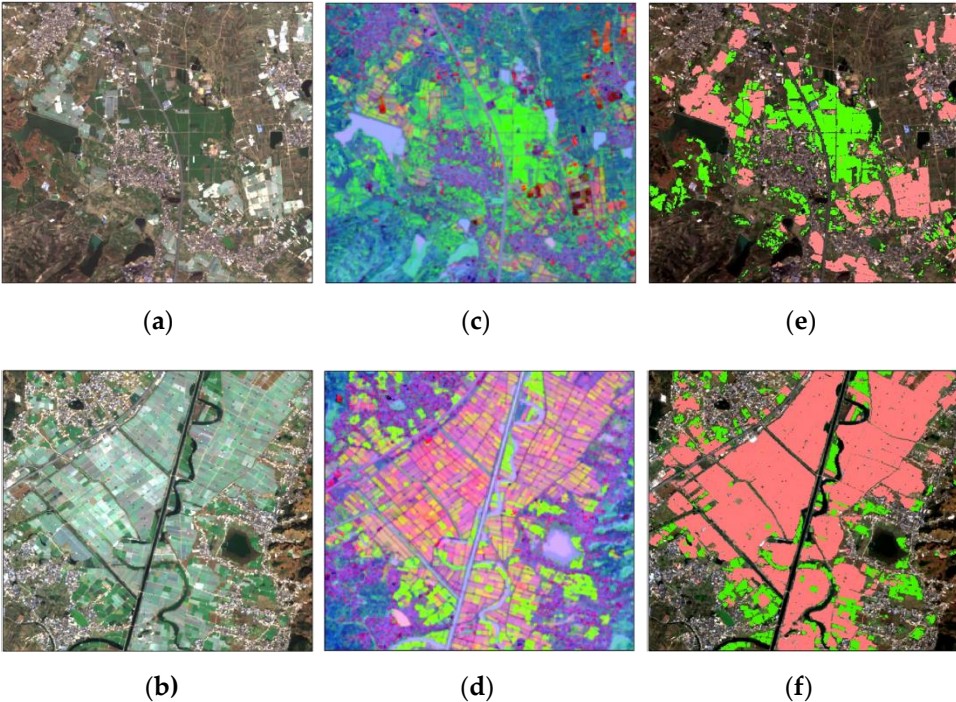

(a)    (c)    (e)

(b)    (d)    (f)

**Figure 8.** Comparison of classification results in Luliang County. (**a**,**b**) display two S2 images (bands: RED/GREEN/BLUE); (**c**,**d**) shows the corresponding visualization results of principal components combined with S1 and S2 band information; (**e**,**f**) shows the mapping results of rice and agricultural greenhouses in Scenario F10 with a spatial resolution of 10 m.

## 4. Discussion

### 4.1. The Reliability of the Generated High-Resolution Planting Structure Map

Yunnan is characterized by cloudy and rainy weather, the availability of optical images is few, so it is difficult to form effective temporal spectral data, and there are certain limitations in the extraction of crop critical growth periods. Conventional crop classification methods based on optical data are difficult to obtain high-precision spatial distribution maps of crop planting structures. In this study, based on the GEE platform combined with Sentinel active and passive remote sensing data and combined with rice phenology images, the median value was synthesized to obtain an effective image of the study area excluding

cloud disturbance. On the premise of making full use of band information of optical and polarization radar data, the PCA-MR method is constructed. Finally, different feature spaces are combined to obtain the best classification scenario and optimal mapping results (Figure 9). The successful implementation of our study is attributed to the availability of high-resolution remote sensing data, abundant and accurate ground sample data, the most suitable extraction method for the region of broken plots and complex planting structures, the best feature scene design, stable algorithm, and GEE platform.

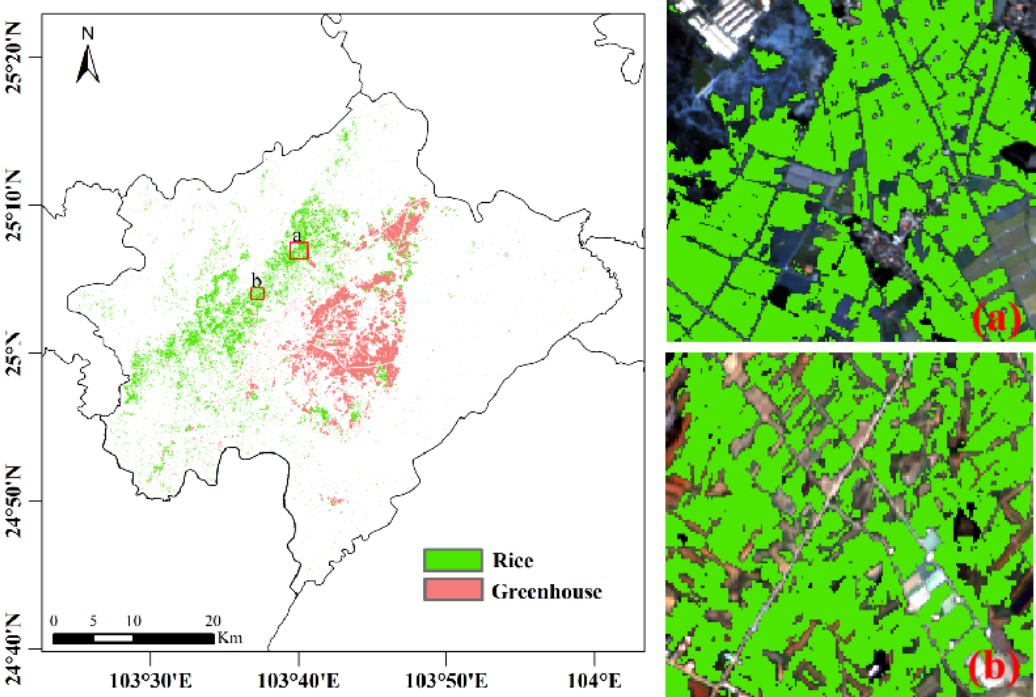

**Figure 9.** The spatial distribution map of rice and agricultural greenhouses in the optimal classification scenario F10 (**a**,**b**) represent two case results of local rice classification.

The integration of Sentinel-1 and Sentinel-2 data provides more possibilities for rice and agricultural greenhouses mapping. We studied the effects of different features on mapping the spatial distribution of rice and agricultural greenhouses (Table 3). Repeated experiments in different scenarios showed that: (1) comparing F1–F2 and F3–F5, we found that Sentinel-2 was more suitable for rice and agricultural greenhouses extraction than Sentinel-1; (2) F8 and F11 classification results are not fully expressed because S2 texture features involved in classification do not preferably improve the final results, so the details of broken steps are not obvious; (3) the accuracy of F6–F8 is significantly higher than that of F1-F5, indicating that it is easier to obtain high-precision mapping results by combining S1 and S2 data compared with a single data source. The final results show that there are certain limitations to using a single image source for crop mapping in complex environments. PCA-MR features are synthesized with S1 and S2 data bands information, which highlights the feature recognition by reducing data redundancy, and provides more possibilities for improving the effect of large-scale mapping classification.

The study area is characterized by typical plateau terrain, landform, and climate conditions, and most of the cultivated land is presented in the form of scattered, broken, and small areas. Potential challenges for taxonomic studies in this region include the availability of key phenological period data for target crops, the availability of ground sample data, the impact of mixed pixels, and the impact of complex crop planting structure, forcing the study area to become one of the most difficult areas for rice and agricultural greenhouses mapping. Therefore, theoretically, as long as the phenological information of

the target crop is obtained, the spatial distribution map of the target crop in any region of the world can be quickly obtained by using this research method.

### 4.2. Application of PCA-MR to Large-Area Mapping

Numerous studies have shown that the calculation of index features from spectral information is effective for rice classification. For example, He et al. analyzed the relationship between EVI, LSWI, and NDVI for mapping rice in southern China [21]. However, influenced by the topography and climate of southern China, resulting in fragmented arable land and complex cropping structures, these indices, although they can play a positive role in classification, are not universally applicable.

To improve this problem, PCA-MR made the first attempt to combine band information from optical and radar data, perform band preference based on GINI coefficients, and perform principal component analysis on the optimized 12 bands to obtain three features. As shown in Table 7, the results show that relative to the classical classification scenario F7, all five categories of PA have been improved, except for the PA of the forest, which remains unchanged, and the impervious surface has been improved the most with 1.99%. These three features not only improve the separability of ground objects but also have a positive impact on the final mapping accuracy. Although PCA-MR has achieved satisfactory results in the classification of rice and agricultural greenhouses, there are still some noteworthy shortcomings. First, PCA-MR requires a combination of optical and radar data analysis. In the case of an extreme lack of optical data, only using radar data is poor. Secondly, deep learning brings new development to remote sensing. Compared with traditional machine learning, the DL model has stronger learning ability. However, the current method of combining DL and GEE is not mature enough to be applied to large-area mapping. Finally, the total classification accuracy is about 0.7 when only three features of PCA-MR are used for classification, and these features can improve the classification effect and accuracy but are not ideal when used as the main classification. Overall, we hope that PCA-MR can promote the synthesis of remote sensing features from single as well as multi-source data, develop more useful features, and provide new paradigms for accurate agricultural mapping rather than being limited to existing methods and known features.

**Table 7.** The optimal classification results of PCA-MR and optical and radar data fusion (F10).

| Reference Class | Rice | GH | Water | Forest | Imp | Others | Total | PA (%) | OE (%) | PA for F7 (%) |
|---|---|---|---|---|---|---|---|---|---|---|
| Rice | 626 | 0 | 0 | 2 | 0 | 15 | 643 | 97.36 | 2.64 | 97.05 |
| Greenhouses | 0 | 541 | 0 | 0 | 12 | 8 | 561 | 96.43 | 3.57 | 96.08 |
| Water | 0 | 0 | 52 | 1 | 0 | 1 | 54 | 96.30 | 3.70 | 96.29 |
| Forest | 1 | 0 | 0 | 225 | 0 | 7 | 233 | 96.57 | 3.43 | 96.57 |
| Impervious | 10 | 11 | 0 | 0 | 538 | 46 | 605 | 88.93 | 11.07 | 86.94 |
| Others | 31 | 11 | 0 | 5 | 28 | 678 | 753 | 90.04 | 9.96 | 88.18 |
| Total | 668 | 565 | 52 | 232 | 578 | 754 | 2849 | | OA (%): 93.47 | |
| UA (%) | 93.71 | 96.09 | 100.00 | 96.57 | 93.08 | 89.80 | | | Kappa: 0.92 | |
| CE (%) | 6.29 | 3.91 | 0.00 | 3.43 | 6.29 | 10.20 | | | | |

UA: user's accuracy, OA: overall accuracy, PA: producer's accuracy, GH: greenhouse, Imp: impervious, OE: omission error, CE: commission error, PA for F7 represents the PA value of classification scenario F7.

### 4.3. Uncertainty

Although the classification results for rice and agricultural greenhouses were shown to be reliable in the final mapping results, there were some uncertainties. First, the availability of S2 data may affect the classification to a certain extent due to the geographical location and climatic conditions. Although the de-clouding process can be accomplished on the GEE platform, there are still limitations in some cloudy areas or in the case of a severe lack of optical data [40–42]. Therefore, how to accurately deal with clouds is worth further research. Secondly, PCA-MR can improve the problem of "different spectrum of the same object, foreign objects in the same spectrum", but it cannot completely remove them. These effects

are still the main problems pixel classification faces. As the study area is located in the plateau region, compared with the plain region with little topographic fluctuation, the plot is more fragmented, the planting structure is more complicated, and most of the cultivated land is dispersed and not large-scale, which is also the main reason for the generation of mixed pixels. In addition, this study only considered the remote sensing classification based on pixels, and the object-oriented method combined with the RF algorithm provided a new idea for the extraction of land cover types [43–45]. How to gradually improve the mapping effect of rice and agricultural greenhouses by incorporating different data sources and advanced algorithms is an important research direction. Thirdly, single-cropping rice is planted in the study area all year round, and it does not take phenological information into consideration when planning agricultural greenhouses. Therefore, the period from July to November, which is from rice planting to harvest, is used for the image median composite, but the different effects of early rice, middle rice, and late rice in this study classification are ignored. One of our future research directions is to further explore the effects of different crop growth characteristics, different classification methods, different data sources, and different remote sensing parameters on remote sensing mapping.

## 5. Conclusions

Based on the GEE platform, time series S1 and S2 images, rich ground sample truth values, and classification feature space of key phenological periods, we proposed a classification method incorporating principal component information and active and passive remote sensing data. The PCA-MR method was proposed to maximize the separability of rice, agricultural greenhouses, and other ground objects and to reduce the errors and impacts caused by pixel classification. The final mapping accuracy is more than 93%, and the range of PU and UA is between 0.94 and 0.97. According to the Yunnan Statistical Yearbook 2020, the rice planting area in the study area is 6300 hectares, and the statistical result of our experiment is 6832 hectares, with a total accuracy of 91.56%. Rice is distributed in areas with sufficient water resources, convenient transportation, and far away from the main urban area. Agricultural greenhouses are distributed near the main urban area, and the distribution of rice is in two directions, with less cross-distribution. This may be because there is less pollution away from the main urban areas, which is conducive to the sowing, growth, and yield increase in rice. Agricultural greenhouses near the main urban areas are mostly for planting seasonal vegetables, which is convenient to quickly supply the daily needs of residents. Both cartographic results and point-based verification prove that our method is successful and has high potential in large-scale cartographic applications. We demonstrate that the overall accuracy of the classification with added texture features is not significantly different from that without added texture features when temporal SAR data is combined with optical data. The proposed cooperative feature synthesis method (PCA-MR) for multi-source remote sensing data can effectively improve the classification effect and accuracy, providing a new idea for precision agricultural mapping.

**Author Contributions:** Conceptualization, visualization, investigation, methodology, writing—original draft preparation, T.Z.; validation, conceptualization, writing—review and editing, B.-H.T.; funding acquisition, supervision, validation, and review, L.H.; formal analysis, writing—review and editing, G.C. All authors have read and agreed to the published version of the manuscript.

**Funding:** This research was funded by the Platform Construction Project of High-Level Talent in KUST. It was also supported by Yunnan Fundamental Research Projects under Grant Nos. 202201AT070164 and 202101AU070161.

**Data Availability Statement:** Sentinel-1 and Sentinel-2A/B data are openly available via the Google Earth Engine.

**Acknowledgments:** We acknowledge the research environment provided by the Faculty of Land Resource Engineering of Kunming University of Science and Technology.

**Conflicts of Interest:** The authors declare no conflict of interest.

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
