# Peer review of "Rice and Greenhouse Identification in Plateau Areas Incorporating Sentinel-1/2 Optical and Radar Remote Sensing Data from Google Earth Engine"

_remotesensing, doi:10.3390/rs14225727_

Round 1

Reviewer 1 Report

This paper presents a new methodology, called PCA-MR, based on Principal Component Analysis (PCA) and GINI Features Selection, for land cover classification using both Sentinel-1 and Sentinel 2 data. The object of the study is the Luliang County in eastern Yunnan Province of China. Emphasis is given in detection of rice plantation and greehouses. Sentinel 1 and Sentinel 2 data fusion is performed using Google Earth Engine. Firstly, eight scenario using, each, a different subset of Sentinel-1 and Sentinel-2 data are tested for land cover classification. Secondly, scenario using PSA-MR in different configurations are tested. The classification efficiency of the different scenario is compared. Although this paper present very interesting results about land cover classification, it suffers from major faults, that should be corrected before publication.

Please find below, the list of comments:

1)    Introduction

a)    In the introduction, there is a lack of clear positioning as regard to previous literature studies performing Sentinel 1 and Sentinel 2 data fusion for land cover classification improvement. Especially, authors should precise clearly their positioning as regard to the references [39-42] of their proposed paper. They should also add the above-mentioned reference treating of classification of land cover in rice, water and bare soil using both Sentinel 1 and Sentinel 2 data: 

Gargiulo M, Dell'Aglio DAG, Iodice A, Riccio D, Ruello G., “Integration of Sentinel-1 and Sentinel-2 Data for Land Cover Mapping Using W-Net”, Sensors (Basel), 20(10):2969, 2020.

doi: 10.3390/s20102969

b)    line 59 : Please define the acronyme “GEE”

2)    Materials and Methods

a)    The paragraph 2.3.3 entitled “Features Selection” is not presenting features selection aspects, but it is describing the construction of features (as for example the contrast feature) from Sentinel 1 and Sentinel 2 data. Therefore, this paragraph should be renamed in “feature establishment” or in a synonymous expression. Moreover, this paragraph should be placed before the paragraph “2.3.2. Principal component analysis of multi-source remote sensing data (PCA-MR)”. Indeed, PCA-MR uses the features build in paragraph currently numbered 2.3.3.

b)    The authors should clarify if the classification is performed in 3, 5 or 6 classes. In lines 92-93, in figure 2 and in line 452, the authors indicate three classes (rice, greenhouse and others). However, in lines 150-153 and in figure 4, the authors indicate 6 classes. In Table 4 and Table 5, the authors indicate 5 classes.

c)    lines 153-157 : “Samples are selected and screened point by point using Google Earth images […] with a total of 5366 samples”. The authors should precise how the class-labelling of each pixel has been performed. In particular, they should precise if the labelling in one of the seven cited classes (rice, agricultural greenhouse, impervious water surface, water, forest land and others) is provided by the data platforms Earth Big Data Sharing Service Platform or Google Earth Engine.

d)    line 147. “Combined […] by the authorities”. This sentence is grammatically incorrect. The authors should revise it.

e)    line 154. The authors should precise what do they mean with the expression “principal component visualized images”

f)     line 180 : “In order to improve the classification effect of rice and agricultural greenhouses and…”. The authors should replace the word “effect” by “efficiency” or a synonymus.

3)    Discussion

The authors should add a clear comparison between classification efficiency obtained without and with PCA-MR. This can be achieved by adding, in table 7, a column that recalls the PA (%) obtained for each class in the case of the best scenario not using PCA-MR. Also, a short paragraph precising for which class PCA-MR allows to obtain a significantly higher PA than with classical method should be added.    

Author Response

请参阅附件。

Reviewer 2 Report

I am rather confused by this work. What is the purpose? What is the main contribution? I hope the authors could clarify the following points:

Motivation

It is well known that SAR and optical images contain land complementary information, and if properly combined, the land classification performance could be improved. Hence the authors’ conclusion that combining S1 and S2 features improves the classification is hardly new.

In the introduction, as the authors said, using optical image for land use classification is a mature practice in the field, the key is cloud problem. In this work, a median image is synthesized from month-long observations, which generally, I believe, could alleviate the cloud problem in a statistical sense. Hence whether the classification could be improved in essence depends on whether the synthesized optical image is less cloudy. If the authors want to emphasize this fact, more experiments should be reported on cloud reduction, rather than on land classification. My view is that, if the cloud could be effectively reduced, the classification performance will increase. Otherwise, it will not. Hence I thought the lengthy presentation on feature extraction, feature combination, and classification seems not quite relevant.

PCS-MR

I cannot grasp its rationality. SAR captures scatterers’ reflection and optical image are for spectral distribution. PCA is to find principal components such that the sum of squared projected errors long such orthogonal directions are ranked from large to small. If the SAR and optical images are decomposed by PCA together, what is the meaning of “the sum of projected errors” in this case? I cannot follow the logic. The combined SAR and optical images is a manifold, what is the reason of summing up errors of drastically different nature?  I am rather confused by such an undertaking, claimed by the authors as a key original contribution.

Besides the above points, the presentation and organization should also be improved. For example, in the introduction, many irrelevant parts are included. For example, I do not know that is the relevance to connecting BRI?     

In “ excavating the advantages and functions of PCA-MR”, “excavate” is not a correct word, “ excavate” basically means to dig a hole. I suppose “ take the advantage ” could be fine.

All in all, I hope the manuscript should be substantially revised, in particular, first clearly state your chief motivation and report the results to support the motivation, and other non-essential results should be removed. And secondly explain the rationality of computing and using PCA-MR features. Of course, introduction should be focused and reduced also. The current manuscript seems unnecessarily long

Reviewer 3 Report

This paper combines sentinel-2 optical with Sentinel-1 radar data to conduct crop classification for plateau areas. The authors developed a principal component feature synthesis method to improve classification accuracy. The authors set 11 scenarios to compare the classification effect of the PCA-MR method. Review The paper is interesting and, in general, well-written. The system is clearly described and the experiments are convincing. For these reasons I recommend the paper for publication provided that a few minor changes are performed.

1.     The introduction section is not enough. Some studies combined optical with radar data to classify crops or other categories. You need to do some review in the Introduction section and have some analysis in the Discussion section of your method compared with the previous methods.

2.     For rice classification, many studies use transplanting period images to distinguish other crops. Could have a comparison with the article’s method.

3.     What is the performance of only using PCA-MR features, and which accuracy could get?

4.     For line 186, how many bands used to be the input of PCA? How to determine the threshold value of Gini?

5.     How to guarantee the accuracy of the 5366 samples obtained by visual interpretation?

Round 2

Reviewer 1 Report

The authors performed corrections that satisfactorily answer to all the comments. The work is interesting and should be published.